# Funneling Spontaneous Emission into Waveguides via Epsilon-Near-Zero Metamaterials

**DOI:** 10.3390/nano11061410

**Published:** 2021-05-27

**Authors:** M. Channab, C. F. Pirri, A. Angelini

**Affiliations:** 1Dipartimento di Scienza Applicata e Tecnologia, Politecnico di Torino, C.so Duca degli Abruzzi 24, 10129 Turin, Italy; fabrizio.pirri@polito.it; 2Advanced Materials and Life Sciences, Istituto Nazionale di Ricerca Metrologica (INRiM), Strada delle Cacce 91, 10135 Turin, Italy

**Keywords:** epsilon-near-zero metamaterial, photonics devices, integrated optics, spontaneous emission

## Abstract

In this work, we discuss the use of epsilon-near-zero (ENZ) metamaterials to efficiently couple light radiated by a dipolar source to an in-plane waveguide. We exploit both enhanced and directional emission provided by ENZ metamaterials to optimize the injection of light into the waveguide by tuning the metal fill factor. We show that a net increase in intensity injected into the waveguide with respect to the total power radiated by the isolated dipole can be achieved in experimentally feasible conditions. We think the proposed system may open up new opportunities for several optical applications and integrated technologies, especially for those limited by outcoupling efficiency and emission rate.

## 1. Introduction

Photonic devices show intriguing properties and potential for different applications, from sensing to quantum communication [1,2]. By exploiting their particular properties, it may be possible to achieve results that were unthinkable just a few decades ago concerning computational power, communication, and sensing applications. For the development of new photonic devices, metamaterials (MM) have been deeply studied in recent years. Among metamaterials, highly anisotropic metamaterials show appealing and exotic characteristics in their optical behavior, which is ultimately due to their subwavelength structuration. MMs are typically composed of a unitary cell, the meta-atom, whose polarizability defines the optical response. By tuning the geometrical parameters of the meta-atom, it is possible to shape the isofrequency surface, and to consequently engineer the density of optical states (DOS). Through DOS modification, spontaneous emission can therefore be controlled both in terms of decay time and directivity of emission [3,4,5]. Directional emission can be addressed by considering that the Poynting vector is always perpendicular to the isofrequency surface [6]. Engineering spontaneous emission opens up the opportunity to overcome the limitations of photon sources such as their emission rate, the broadened emission spectrum, and directivity [7,8,9].

The efficient coupling of point-like light sources is crucial for several applications, from integrated sensing devices to on-chip quantum information devices. It has been experimentally demonstrated that direct coupling of quantum dots (QDs) with photonic waveguides can reach efficiencies as high as 90% compared with the emission of uncoupled QDs. [10] However, in the same work, the authors point out that efficiency can vary from less than 10% to 90% depending on the emitter position and frequency. Here, we investigate the direct coupling of enhanced directional emission into an in-plane waveguide. A key feature of this system is that it amplifies and redirects light simultaneously, allowing for a global efficiency of greater than 100% to be reached when the uncoupled dipole is taken as a reference.

In detail, we present a theoretical analysis of the injection of spontaneous emission into a waveguide to extract the large k-modes buried in a multilayer MM. Since our final aim was to confine the enhanced spontaneous emission in a waveguide for the aforementioned photonic applications, we focused our attention on directional emission control. 

The optical response of a metamaterial depends on several parameters. In the case of hybrid metal-dielectric meta-atoms, one of the parameters that can be used to control the optical response is the metal fill factor, as can be observed in the following equations where the effective medium approximation (EMA) is used to estimate the components of the permittivity tensor [11]: (1)$$\rho =\frac{d_{m}}{d_{m}+d_{d}}\;,\;\;\;\;\;\;\;\;\;\;\;\;\;\;\;\;\;\varepsilon_{\left|\right|}=\rho \varepsilon_{m}+\left(1-\rho \right)\varepsilon_{d},\varepsilon_{⊥}={\left(\frac{\rho }{\varepsilon_{m}}+\frac{1-\rho }{\varepsilon_{d}}\right)}^{-1}$$
(2)$$\varepsilon =\left(\begin{array}{ccc} \varepsilon_{\left|\right|} & 0 & 0\\ 0 & \varepsilon_{⊥} & 0\\ 0 & 0 & \varepsilon_{\left|\right|} \end{array}\right),{\left(\frac{k_{\left|\right|}}{k_{0}}\right)}^{2}\frac{1}{\varepsilon_{⊥}}+{\left(\frac{k_{⊥}}{k_{0}}\right)}^{2}\frac{1}{\varepsilon_{\left|\right|}}=1$$
where *d_m_* and *d_d_* are the metal and dielectric thicknesses in the unitary cell, respectively; ρ is the metal fill factor; $k_{i}$ and $\varepsilon_{i}$ are the wavevector and electric permittivity components, respectively; and I = ||,⊥ are the directions parallel and perpendicular to the MM surface, respectively. In particular, it was reported in the literature that the emission directivity of a dipole in close proximity to a MM can be controlled by tuning the metal fill factor, and when ENZ condition is approached, vertical or horizontal emission can be achieved [6]. In the EMA, the effective permittivity values determine the curvature of the isofrequency curves (see Equations (1) and (2)). In Equation (3), it is shown that the necessary condition to obtain the horizontal emission for the studied multilayer MM at a fixed wavelength is as follows: (3)$$Re\{|\varepsilon_{\left|\right|}\left(\rho \right)|\}\sim 0^{+}\;\cap^{\;}\;\;|\varepsilon_{\left|\right|}\left(\rho \right)|=o\left(\varepsilon_{⊥}\left(\rho \right)\;\right)$$
where the real part of $\left|\varepsilon_{\left|\right|}\left(\rho \right)\right|$ must be around zero, and $|\varepsilon_{\left|\right|}\left(\rho \right)|=o\left(\varepsilon_{⊥}\left(\rho \right)\;\right)$ means that $\underset{x\to \rho }{lim}\frac{\left|\varepsilon_{\left|\right|}\left(x\right)\right|}{\varepsilon_{⊥}\left(x\right)}=0$. In this paper, we show how effective photon extraction can be achieved with a high level of performance by means of a system composed of a multilayer MM (composed of Au and Al_2_O_x_ layers) and an in-plane waveguide. To this end, we performed the following steps, detailed in the Results section: we determined the fill factor needed for the horizontal emission, modal analysis, and calculation of the dispersion relation of the propagating modes in the MM slab and the waveguide coupling analysis.

## 2. Materials and Methods

All the results reported in this paper were obtained through finite element method (FEM) simulation software, Comsol Multiphysics 5.3. The refractive indexes of the single layers were extrapolated by data acquired with an ellipsometer (J.A. Wollam alpha-SE model) on two samples of 20 nm thickness, separately, with an Au and Al_2_O_x_ layer. In the following sections, we report the methodology used for each Results subsection.

### 2.1. Methodology for Preliminary Results

This first step determined the necessary fill factor to obtain the horizontal emission by comparing the values with the condition reported in Equation (3). For this analysis, we performed 2D geometry simulations. We designed a circular domain with a 5 μm radius, in which we initially placed the MM slab and surrounded it by a perfectly matched layer domain (2 μm thickness). The results were obtained using a MM 3 μm long and 0.2 μm thick, exploiting both the EMA and actual structures. To investigate the effect of the last layer, we modelled the actual structures with two different configurations: in one case, the top layer was composed of Au; in the other, it was composed of Al_2_O_x_. The structure with the Al_2_O_x_ top layer was obtained by adding an Al_2_O_x_ layer (total thickness of 233.2 nm). An electric dipole with a momentum of P = 0.5 mA∙m placed 5 nm above the MM and vertically oriented with respect to the MM surface was used as a photon source. 

### 2.2. Methodology for Modal Analysis

We performed the modal analysis of the MM slab by using a rectangular unitary cell that was 2 μm in height and 0.5 μm in width. The MM was infinitely wide due to the applied continuity periodic boundary conditions. The MM thicknesses were the same ones exploited in the Preliminary Results section. Moreover, the multilayer was placed above a glass substrate with a refractive index equal to 1.45.

### 2.3. Methodology for Performance Analysis

Performance analysis determined which configuration provided the highest performance in terms of power transport and enhancement. The performance analysis was carried out with the same geometry used in the Preliminary Results section: radius of the circular domain, MM dimensions, and dipole orientation, position, and momentum. The Purcell factor is defined as the ratio between the total emitted power by a dipole placed 5 nm above the MM structure and the power radiated by a dipole placed in the air. The lateral emitted power is defined as the line integral of the Poynting vector along a semi-circumference (radius 450 nm) placed 500 nm away from the edge of the MM (Figure 1a). The total power was computed by integrating the Poynting vector along a closed line (a square with a lateral size of 5 nm) containing the dipole, as shown in Figure 1b. 

### 2.4. Methodology for Waveguide Coupling Analysis

For the waveguide, we used a thin layer of dielectric material (refractive index of 2.25) placed above the glass substrate and in contact with the MM. The waveguide length was 8.5 μm, while the thickness was kept equal to that of the MM slab. The external domain and dipole parameters were identical to those mentioned in the Preliminary Results section. The outgoing power from the waveguide was estimated by evaluating the linear integral of the Poynting vector on the vertical line that delimited the waveguide’s end. We define global efficiency as the percentage ratio between the power exiting from the waveguide outlet and the total power emitted by an isolated dipole in the air. In this way, we estimated the performance of the proposed global structure. To evaluate the coupling efficiency between the dipole and the waveguide mediated by the MM, we performed a parametric study over the MM length.

## 3. Results 

### 3.1. Preliminary Results

We investigated the multilayer structure using the EMA to find the optimal condition for lateral emission. In particular, as suggested by Ferrari et al. [6], we fixed a particular wavelength (λ = 637 nm) in order to estimate the permittivity components’ dependence on the fill factor. We fixed this wavelength since it corresponds to the zero phonon line of nitrogen-vacancy centers in diamonds, a solid-state photon source that has been widely investigated in recent years in the field of quantum information technologies and integrated photon sources [12,13,14].

In Figure 2a, the real part of the permittivity components at 637 nm as a function of the metal fill factor is reported. The necessary condition for horizontal emission was found for a fill factor around 0.2 (Figure 2b). For determining the optimal fill factor in which this condition is achieved, we performed a FEM parametric study (see the Methods section) over the fill factor value, by varying it between 0.15 and 0.25, with 0.01 steps.

From this simulation, we found that the optimal condition for horizontal power emission is reached with a fill factor equal to 0.17 (*ε*_⊥_ = 3.508 + 0.058i and *ε*_||_ = 0.032 − 0.006i). These values led to the isofrequency curve reported in Figure 2c. As can be seen in Figure 2d, for optimized horizontal emission conditions, almost all the power flow is horizontal. It is important to note that for the actual metamaterial, this directional emission can be theoretically obtained with different thicknesses of the unitary cell (*a = d_m_ + d_d_*) maintaining the condition *a << λ*. Additionally, the effective medium does not take into account which layer interfaces with the emitter [15]. For these reasons, we compared the EMA and the actual metamaterial in two distinct configurations: a multilayer terminated with a gold layer and another one terminated with alumina. This analysis is reported in the following sections. 

### 3.2. Modal Analysis

In order to determine the out-of-plane wavevector, the propagation length, and the field distribution of the propagating modes that can be injected in a waveguide, we performed a modal analysis in a wavelength region (600–700 nm) around the wavelength of interest λ. The results are reported in Figure 3.

First, we considered the configuration with the Au top layer. The analysis revealed a propagating mode well beyond the glass line (Figure 3a). In Figure 3b, we show its related electric field profile, which was tightly confined inside the MM slab. Adding an Al_2_O_x_ layer resulted in a red-shift of the mode (Figure 3c). Even if the propagating mode shows a similar electric field profile, there is a significant difference in the propagation lengths between the two simulated configurations. For the Au top layer configuration, we found a propagation length ranging from 450 to 300 nm, while for the Al_2_O_x_ top layer, the propagation length was around 200 nm. Interestingly, for the Al_2_O_x_ top layer configuration, an additional mode was found immediately beyond the glass line. This mode is associated with lower wavenumber values and about two orders of magnitude larger propagation lengths with respect to the modes discussed above. In Figure 3d, its electric field profile is reported. As shown, this mode is weakly confined. Both modes are TM−polarized. 

### 3.3. Performance Analysis

To determine the optimal configuration, both the Purcell factor and the percentage of laterally flowing power radiated by a dipolar source placed on top of the MM were estimated. 

Figure 4a shows the field distribution at λ = 637 nm in the two configurations. As shown in Figure 4a, lateral emission occurred in both configurations, although the field distribution looks rather different. By analyzing the Purcell factor (Figure 4b), it is clear that the Au top layer provided a higher value of the Purcell factor with respect to the Al_2_O_x_ top layer over the entire wavelength range considered. This behavior is due to the coupling of the dipole with plasmonic waves sustained by the thin layer of gold [16,17,18], which was less efficient in the second configuration. This coupling allows reaching a Purcell factor as high as 59 at 540 nm. Moreover, observing Figure 4c, it seems that the Al_2_O_x_ top layer provided a better lateral emission since the percentage of lateral emitted power reached peaks of 58% of the total radiated power. The field distributions and the performance analysis suggest that the choice of the best configuration is not unambiguous; as the Au top layer configuration provides a larger Purcell factor but shorter propagation lengths, the overall length of the MM (or alternatively, the distance of the emitter from the MM lateral edge) plays a crucial role. We studied the coupling of laterally propagating radiation with a waveguide exploiting the MM with the Au top layer, as it is the configuration that provides the largest enhancement in the overall emission. 

### 3.4. Waveguide Coupling Analysis 

We analyzed the coupling of the emitter with an in-plane waveguide mediated by the MM. The field distribution in Figure 5a shows that the radiated field naturally couples to the waveguide modes with relatively low scattering. Since in the multilayer MM the emission can be affected by the losses in the Au layers, we studied the outgoing power dependence on the length of the MM structure. In Figure 5b, we show the percentage ratio between the outgoing power and the total amount of emitted power by the dipole. The calculated behavior is well-fitted by the exponential function (f(x) = ae^bx^) where a=4.81±0.07, and $b=\left(-1.68\pm 0.05\right)\;{\mu m}^{-1}.$

In Figure 5b, the percentage values are around 0.5% for lengths of a few microns. Even though these values seem very low compared with other systems [19], we have to consider the enhancement provided by the MM. We define global efficiency as the efficiency of the emitter above the structure compared with the emitter in the air. In Figure 5c, we show the estimated global efficiencies as a function of the emission wavelength. It is straightforward that the lateral power flow is larger than the power radiated in the air (black dashed line) used as a reference in a broadband region. Significantly high peaks were reached for specific wavelengths, for example, 1414% for 630 nm using a 150 nm long MM and 377% for 632 nm using a 500 nm long MM. Another important feature presented in Figure 5c is that for a MM longer than 500 nm, a valley can be observed between wavelengths from 610 to 625 nm, while it is not present in the 150 nm long MM. This feature can be explained by considering the propagation lengths of the modes found in the modal analysis subsection, in which we reported propagation lengths below 500 nm in the same wavelength region. For this reason, there is additional power that can be injected into the waveguide when the MM length is below 500 nm. In Figure 5d, we report the MM length dependence of the global efficiency integrated on the highlighted region in Figure 4c (550–700 nm). It was found that percentages larger than 100% can be reached even when considering the integrated values when the MM slab is shorter than approximately 1 μm.

## 4. Discussion

In this work, we studied the coupling of a dipolar source with an in-plane waveguide mediated by a hybrid metal-dielectric metamaterial. Specifically, we demonstrated that the proposed MM can support a horizontal emission in a broadband region with propagating large k-modes. Moreover, it is characterized by significant Purcell factor values. Numerical models show that lateral emission can be easily injected into a waveguide to extract the large k-modes. Importantly, the proposed system can sensitively increase the number of photons that can be extracted from the light source, especially when a single wavelength is considered. This feature makes the system appealing for integrated photonic applications in which photon extraction is a crucial parameter, especially when a single wavelength is considered. Our analysis shows that the global efficiency increases exponentially with decreasing MM length due to ohmic losses, and high performance can be achieved when the distance between the emitter and the waveguide is shorter than one micron.

## Figures and Tables

**Figure 1 nanomaterials-11-01410-f001:**
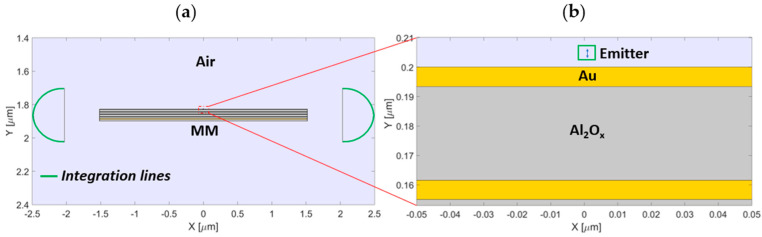
(**a**) Geometrical scheme exploited for the performance analysis. (**b**) Magnification of the red−dotted region highlighted in (**a**), showing the dipole and the square along which we calculated the total power.

**Figure 2 nanomaterials-11-01410-f002:**
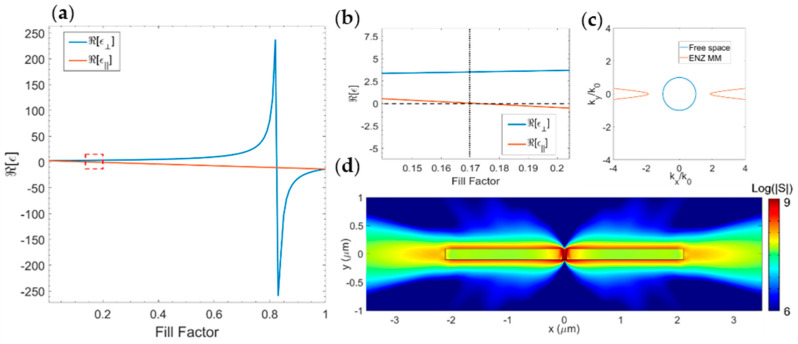
(**a**) Real part of the permittivity components as a function of the metal fill factor in the effective medium approximation. (**b**) Magnification of the red−dotted region highlighted in Figure 1a, showing the region in which the horizontal emission is expected. (**c**) Isofrequency curves for the air (blue circle) and the ENZ metamaterial (red line) resulting from Equation (3) when *ε*_⊥_ = 3.508 + 0.058i and *ε*_||_ = 0.032 − 0.006i. (**d**) The magnitude of the Poynting vector plotted for the effective medium with a fill factor equal to 0.17 at *λ* = 637 nm.

**Figure 3 nanomaterials-11-01410-f003:**
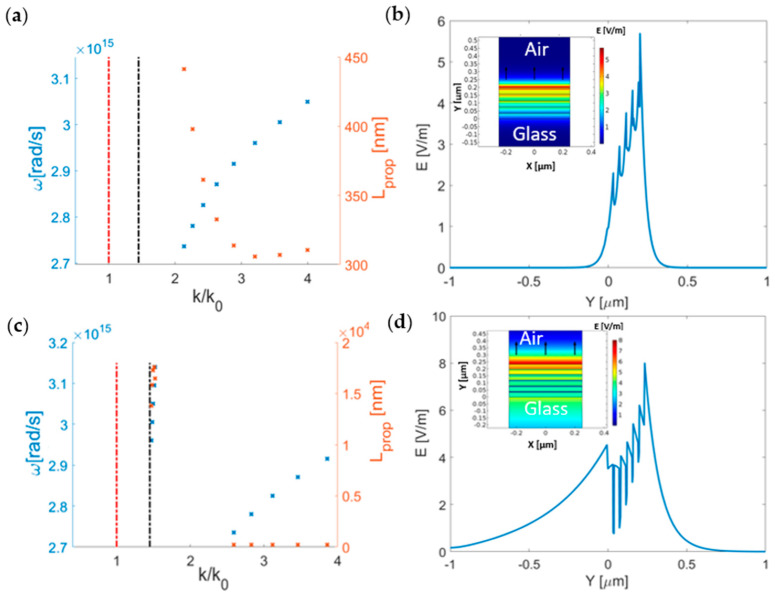
(**a**,**c**) Dispersion relation (blue) and propagation length (red) of the propagating modes found with the modal analysis for the Au and Al_2_O_x_ top layer configurations, respectively. The black and bright red dash−dotted lines show the light lines in glass and air, respectively. (**b**,**d**) Electric field profile at 637 nm (*ω* = 2.96 × 10^15^ rad/s) extracted along the Y direction in the center of the MM with the Au and Al_2_O_x_ top layers for the propagating modes. In the inset, the corresponding surface plots of the E field are shown. The black arrow (electric field orientation) shows that the modes are TM−polarized.

**Figure 4 nanomaterials-11-01410-f004:**
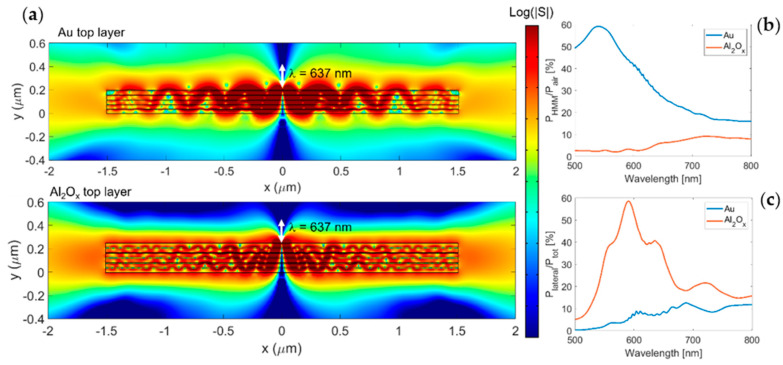
(**a**) Magnitude of Poynting vector (on a logarithmic scale) of the field radiated by a dipolar source (white arrow) vertically oriented placed 5 nm above the MM surface plotted respectively for the ENZ metamaterial with the Au top layer (top) and Al_2_O_x_ top layer (bottom). (**b**) The Purcell factor at different wavelengths was calculated for the Au top layer (blue line) and the Al_2_O_x_ top layer (red line) configurations. (**c**) The percentage of lateral emitted power with respect to the total power is reported for the Au (blue) and Al_2_O_x_ (red) top layer configurations.

**Figure 5 nanomaterials-11-01410-f005:**
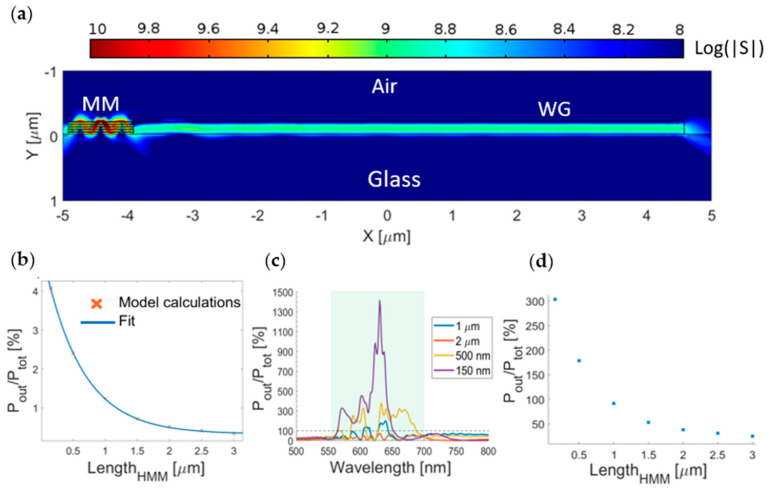
(**a**) Magnitude of the Poynting vector at 637 nm for a 1 μm long MM coupled with the waveguide. (**b**) Percentage ratio between the total emitted and outgoing power with single exponential dependence on the MM length. (**c**) Global efficiency in the wavelength region of interest for different MM lengths. (**d**) Global efficiency integrated over the highlighted region in (**c**) for different MM lengths.

## Data Availability

The data presented in this study supporting the results are available in the main text. Additional data are available on request to the corresponding author.

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
