# Peer review of "Funneling Spontaneous Emission into Waveguides via Epsilon-Near-Zero Metamaterials"

_nanomaterials, 2021, doi:10.3390/nano11061410_

Round 1

Reviewer 1 Report

Theoretical modeling papers are always difficult to evaluate in terms of novelty and significance.  Such papers with corresponding experimental measurements are much preferred.  

Here the authors point out that extraction of light coupled to modes bound to the metamaterial can be an issue.  They go on to present a theoretical analysis of the injection of spontaneous emission via a metamaterial into a waveguide.  However, they fail to compare their results to what has been proposed and demonstrated in terms of extraction in the literature.  If this omission and some minor corrections mentioned in the attached file are addressed the paper could be made suitable for publication.

Author Response

Dear Editor,

we would like to acknowledge the referees for their thorough work while evaluating the manuscript and their very precise and convenient suggestions. In the following letter, we provide an answer to the referees’ comments with a description of the corresponding modifications in the manuscript. We believe that the updated version of the manuscript satisfactorily addresses all the issues raised by the referees and that the manuscript has been improved following the reviewers’ remarks. All the modifications to the main text are highlighted in yellow in the attachement.

Best regards,

Marwan Channab

Answers to referees’ remarks

Reviewer 1

Here the authors point out that extraction of light coupled to modes bound to the metamaterial can be an issue.  They go on to present a theoretical analysis of the injection of spontaneous emission via a metamaterial into a waveguide.  However, they fail to compare their results to what has been proposed and demonstrated in terms of extraction in the literature.  If this omission and some minor corrections mentioned in the attached file are addressed the paper could be made suitable for publication.

We thank the reviewer as he has raised a crucial point and based on his comment we agree that the introduction can be a bit misleading about the purpose of the paper. Indeed, we do not demonstrate an efficient extraction mechanism, but rather we investigate a mechanism to inject light into a waveguide (WG).

We also agree that our result needs to be compared to what is shown in the literature to be meaningful. To address these two points we modified the introduction and added a reference [Laucht A. et al, A Waveguide-Coupled On-Chip Single-Photon Source, Phys. Rev. X, 2, 011014 (2012)] for comparison. In this valuable work, the authors experimentally address the problem of estimating the coupling efficiency, showing that in their system it can be as large as 85-96% in the best case. However, they also mention that for different sources, it can vary from less than 10% to more than 90% depending on the position and frequency of the emitter.

We point out that in this work the efficiency estimation is indirectly made by comparing the lifetime of coupled emitters to the lifetime of uncoupled ones, meaning that it has to be compared to what we defined as “global efficiency”. This parameter is relevant when one is interested in the overall power injected into the waveguide (for example in sensing applications) but it may be misleading when photon statistics is relevant.

We modified the introduction and added the reference according to the above discussion.

Line 64:

It would be helpful for the reader to give a description/interpretation of eq. 3. Also, what is the meaning of the right side of the equation?

As suggested by the reviewer, we have explained the meaning of eq.3 and of the right side of the equation.

Line 79:

This isn't really results but the geometry being modeled.  A figure describing the geometry would be helpful.

We apologize as the subsection title may have caused a misunderstanding. This section does not report the results, but the methodology related to the results sections. To avoid this misunderstanding, we have renamed the Methods subsection “methodology for preliminary results” and so on for each methodology used for the related results section.

Lines 107-110:

This again could use a figure to help the reader better visualize what these definitions mean.

To meet the reviewer suggestion, we added Figure 1 to better visualize the definitions mentioned in section 2.3 and we renamed all the figures consequently.

Line 208:

Did you mean "unambiguous"?

We thank the reviewer for her/his suggestion, we substituted “univoquous” with ”unambiguous” in the text.

Line 226:

Don't call the red x results "Data".  Rather use something like "model calculations".  Data implies that experimental measurements were made, not modeling calculations.

We agree with the reviewer that data implies that experimental measurements were made. We have changed the label in Figure 5b with “Model calculations” as suggested.

Reviewer 2 Report

The authors demonstrate that the coupling to the in-plane waveguide is optimized by utilizing the directional emission by the ENZ metamaterial. The basic concept is reported in Ref [6]. First, the authors optimized the injection of light into the waveguide by adjusting the metal filing factor in the ENZ metamaterial. Assuming a point source at 637 nm, the authors calculated the wavenumber and propagation length for two structure with different top layers. Finally, the authors calucluated the coupling to the waveguide, and they found that large coupling using short ENZ slab (<1 μm). I agree that their simulated results  are reliable and will attract many readers. Therefore, I recommend the publication in Nanomaterials.

Correct  minor error before publication.
line 148 "epsilon_ {//}"

Author Response

Dear Editor,

we would like to acknowledge the referees for their thorough work while evaluating the manuscript and their very precise and convenient suggestions. In the following letter, we provide an answer to the referees’ comments with a description of the corresponding modifications in the manuscript. We believe that the updated version of the manuscript satisfactorily addresses all the issues raised by the referees and that the manuscript has been improved following the reviewers’ remarks. All the modifications to the main text are highlighted in yellow in the attachment.

Best regards,

Marwan Channab

Reviewer 2

The authors demonstrate that the coupling to the in-plane waveguide is optimized by utilizing the directional emission by the ENZ metamaterial. The basic concept is reported in Ref [6]. First, the authors optimized the injection of light into the waveguide by adjusting the metal filing factor in the ENZ metamaterial. Assuming a point source at 637 nm, the authors calculated the wavenumber and propagation length for two structure with different top layers. Finally, the authors calculated the coupling to the waveguide, and they found that large coupling using short ENZ slab (<1 μm). I agree that their simulated results are reliable and will attract many readers. Therefore, I recommend the publication in Nanomaterials.

We thank the reviewer for his comment and appreciate his statement about the quality of the paper.

Line 148:

 "epsilon_ {//}"

We included the suggested correction.

Round 2

Reviewer 1 Report

Thank you for adding Figure 1 and the other suggested changes.  However the figure could use some labeling of the components shown to better describe it.  In the caption, "the square along with we have calculated" should be "the square along which we have calculated".

Author Response

Answers to referees’ remarks

Reviewer 1

Thank you for adding Figure 1 and the other suggested changes.  However the figure could use some labeling of the components shown to better describe it.  In the caption, "the square along with we have calculated" should be "the square along which we have calculated".

We thank the reviewer for her/his suggestion, we have labeled the components in Figure 1 and included the suggested correction. Also in this case we have highlighted the correction in the attachment
